# Photovoice—Towards Engaging and Empowering People with Intellectual Disabilities in Innovation

**DOI:** 10.3390/life10110272

**Published:** 2020-11-05

**Authors:** Sofie Wass, Mugula Chris Safari

**Affiliations:** 1Centre for eHealth, University of Agder, 4879 Grimstad, Norway; 2Department of Psychosocial Health, University of Agder, 4879 Grimstad, Norway; chris.safari@uia.no

**Keywords:** photovoice, intellectual disability, assistive technology, innovation

## Abstract

E-health and welfare technology offer new ways to support health and social care delivery. While initiatives are made to engage disadvantaged user groups in innovation, people with intellectual disabilities tend to be excluded from design activities. This is a concern as this group can benefit from the use of assistive technology. However, it can be time-consuming and challenging to involve end-users in the design of technology. This calls for processes that are creative, empowering and that facilitate user involvement. In this study, we report and reflect on the process of using photovoice to understand user needs and to empower participants with intellectual disability in an innovation process. Nine persons with intellectual disability participated in photovoice to identify user needs connected to the design of assistive technology. The findings in our study suggest that the use of photovoice can contribute to the sharing of contextual and individual needs and an empowerment process that includes coping, self-determination and ownership. Photovoice can be a tool to reduce or remove some of the challenges that are faced when identifying user needs and is a way to strengthen the individual’s capacity to cope with the demands of participating in an innovation process.

## 1. Introduction

It is well recognized that user involvement is important for the design, impact and adoption of technology, both in healthcare and social care [1]. However, it is time-consuming and challenging to involve end-users in design processes [2]. For those living with cognitive disabilities such as intellectual disabilities, this can be especially challenging. Earlier research has highlighted that intellectual disability characteristics may create barriers and limitations when involving this population in design activities [3]. Although these characteristics often vary in different individuals with intellectual disability, there are some difficulties that people with intellectual disabilities are likely to encounter. Characteristics such as communicative difficulties, difficulties with abstract concepts, understanding turn-taking and difficulties expressing themselves adequately may pose challenges during design activities [4,5]. While initiatives have been made to include people with intellectual disabilities in research [6], this group of users tends to be excluded from design activities [7]. Thus, their needs are not included in the design process or merely communicated by people close to them [7]. This is a concern as people with intellectual disabilities can benefit greatly from the use of assistive technology [8]. As highlighted in previous research, this is distressing as it can lead to even more inequalities between advantaged and disadvantaged users of technology if needs are not met or taken for granted [9,10].

The primary rationale for involving users in innovation and design is twofold: to increase system satisfaction and to give a voice and empower future users [1,2]. Photovoice is such a research method where participants are actively involved in the research process [11]. As described by Booth and Booth [12] (p. 431), "photovoice uses photography as a means of accessing other people’s worlds and making those worlds accessible to others". The method was developed by Wang and Burris [13,14] and was traditionally used to conduct participatory needs assessments and evaluations in health promotion and education. In photovoice, the participants take photographs that document various aspects of their lives, which are later used as input in qualitative interviews to encourage reflections on feelings and experiences [14,15]. As a technique, photovoice aims to enable participants to find personal strength and project a vision of their lives that may educate others to better understand their reality. Photovoice is a part of a growing trend in the use of participatory research approaches with people with intellectual disabilities as it can offer participants the opportunity to engage in research [11]. Still, photovoice requires researchers to share power and control [11] and facilitate the process for useful involvement of users.

In this study, we contribute to the understanding of user involvement of vulnerable user groups in the design of assistive technology. The aim of our study is to report and reflect on the process of using photovoice to understand user needs and to empower participants with intellectual disability in an innovation process. More specifically, we aim to answer the following research question: how can photovoice empower participants with intellectual disabilities when participating in an innovation process?

### 1.1. Intellectual Disability

Intellectual developmental disability is defined as “a group of developmental conditions characterized by significant impairment of cognitive functions, which are associated with limitations of learning, adaptive behavior, and skills” [16] (p. 177). The primary descriptors of intellectual disability are a marked impairment of core cognitive functions, which lead to limitations in the development of knowledge, reasoning and symbolic representations. People with intellectual disabilities have difficulties in verbal comprehension and working speed, as well as working memory. Intellectual disability is classified into four main clinical severity subcategories, mild, moderate, severe and profound intellectual disability, in addition to the provisional category of unspecified intellectual disability [16]. Intellectual disability is typically operationalized as scoring more than two standard deviations below the population mean on an intelligence test [17,18] and it is often a lifelong health condition that originates before age 18 [16]. Approximately 1–2% of the world’s population has an intellectual disability and it is predicted that this population will grow [19].

Historically, the concept of disability has been defined by the medical and social model of disability. The medical model, also referred to as the deficit or individual model of disability, views the individual as a patient [20]. This model places the problem or impairment with the individual, meaning that the person must be treated or rehabilitated as far as possible [21]. This view makes disability a technical problem rather than a social one. However, the medical model is usually mentioned in a critical light, with little discussion devoted to defending the model [20]. The social model of disability, which calls for social and structural change, places the problem with society [21]. In contrast to the medical model, the social model of disability emphasizes that social and structural change must be made to enable full participation for people with disabilities [20,22]. 

In recent years, a third perspective on disabilities has emerged: the Nordic relational model, also known as the Scandinavian model and the Gap model. According to this model, disability occurs when there is a mismatch between the individuals’ abilities and the demands from the context or the society [22]. Within the gap model, disabilities may be reduced or removed by either changing the environment or strengthening the individual, or both [21]. However, in recent years, the role of critical disability studies has grown and disability research seems to be shifting towards posthuman approaches to disability [23]. There is also an ongoing development of the theorization of learning disability to provide more inclusive approaches [24].

### 1.2. Empowerment

Empowerment is a concept that incorporates issues of control, awareness and participation [25]. It is, therefore, closely connected to the goals of user involvement and photovoice [11,13,14]. Empowerment can be understood both as a process and as an outcome of such a process. Zimmerman and Warschausky [25] provide a framework for empowerment which includes three dimensions: values, processes and outcomes that all contribute to empowerment. Empowerment values can be described as an approach that guides the collaboration between the person with disabilities and the support network. Empowering processes are those opportunities and actions that support persons to “gain mastery and control over issues that concern them, develop a critical awareness of their environment, and participate in decisions that affect their lives.” [25] (p. 5). In addition, these processes are areas for learning and an opportunity to influence the environment. Empowerment outcomes deal with consequences of being involved in an empowering process [25]. There are several ways and degrees of including and empowering research participants. In design and information systems research, common concepts include co-design, co-creation and participatory research [2,26]. Within disability studies, inclusive research is another example, used to classify research where people with disabilities are actors and co-researchers of projects and academic output [27].

## 2. Materials and Methods 

In this section, we describe the photovoice process and ethical considerations. 

### 2.1. Data Collection

The data collection on user insights included nine photovoice interviews with adults with intellectual disability. The participants were recruited in close collaboration with employees at three different sheltered workplaces in Norway. The employees were asked to nominate potential participants based on their ability to (1) give consent, (2) take photos on their own or with the assistance of a researcher and (3) describe and reflect verbally on photos taken. In addition, they either had to walk or go by bus, taxi or use other means of transportation to get to work. To ensure that informed consent was given, the manager first informed the participants about the aim of the study and the photovoice activity. As a second step, the researchers visited the workplace to present the research project, themselves, how to take photos and with what aim. We also talked about ethics in regard to taking photos as data protection restricted whom the participants were allowed to include in photos without written consent. As a reminder, each participant was given printed instructions on what to focus on when taking photos. To identify user needs for transport support, we used the following instruction: On your way to and from work take photos of things, places or animals that are important, difficult or which make you happy/calm/insecure/sad/scared/stressed. We offered to send a text message reminding the participants to take photos, but all declined.

The participants included seven men and two women with mild or moderate intellectual disability. Five participants traveled by public transport to work (three by bus and two by combining subway and bus), one participant by taxi, two participants by maxitaxi and one participant by combining taxi and bus. All but one participant used their smartphones to take photos on their way home and/or to work. Two participants decided to not take any photos but took part in the interview (Table 1). 

The day after the photos were taken, we conducted interviews with the participants. The interviews took place at their workplaces and the participants could invite a colleague or manager to accompany them during the interview if they wished (no one did so). The focus of the interviews was twofold: (i) to identify user needs for the design of transport support and (ii) to identify the participants’ experiences of using photovoice. During the first part of the interview, the participants were asked to tell us about their current mode of transportation and their experiences connected to traveling. As a second part, the participants were asked to show us the photos they had taken, one by one. In connection to each photo, we asked what the photo showed, why they had taken it, if it represented something important or difficult and how the photo made them feel. They were also asked if they would have liked to have taken other photos. The third part of the interview focused on the possibilities of using other means of transportation. During the final part of the interview, the participants were asked about the use of photovoice as a method. All interviews were recorded and later transcribed by the researchers.

### 2.2. Data Analysis

The data material presented in this study (i.e., participants’ experiences of using photovoice) was analyzed by thematic analysis and followed the steps as recommended by Braun and Clark [28]. The approach was chosen as the method is flexible and helps to identify, analyze and report patterns throughout collected data [28]. To start the analysis of the data, the photos taken were inserted into the interview transcripts. This made it easier to connect the reflections made by the participants to their photos. During the first step, the material was reread several times to gain familiarity with the data. Initial thoughts were also noted in the margin of the interview transcripts. As a second step, the two researchers generated initial codes, focusing on the participants’ experiences of using photovoice. This first coding was done individually by the researchers. The codes and the extracted data were then discussed between the two researchers to reach a consensus. Minor changes were made to the codes based on the discussion. As a third step, the codes were grouped into potential themes. Through a discussion between the two authors, a thematic map was generated (Figure 1).

### 2.3. Ethical Considerations

The participants in the study were informed about the aim of the research project. They were told that participation was voluntary and that they could withdraw from the research project at any time without any consequences. This was stressed when the study was presented, when the photo session started and at the interview. Regulations regarding data protection prevented the participants from taking photos of persons who had not given formal consent to be part of the study. Mitchell [15] discusses ethical issues when doing visual research, including informed consent. In our study, we found it more limiting than valuable to put the responsibility on the participants to interact with unknown people and ask for informed consent. Therefore, we stressed the importance of only taking photos of things, places or animals. The research project was approved by the National Centre for Research Data (648227) and was performed in accordance with the Declaration of Helsinki.

## 3. Results

The thematic analysis resulted in three main themes: (a) experience of coping, (b) sense of self-determination and (c) sense of ownership (Figure 1). 

### 3.1. Experience of Coping

The experience of coping was characterized by three sub-themes: a sense of mastery, environmental challenges and cognitive challenges. Most participants appeared to master the photovoice activity and described it as “fun” and “natural”. During the interviews, the participants were able to describe what the photos represented to them and why they had taken them. Olav described the act of taking photos in the following way: “It was great. It went great to take photos from the bus. It’s not that difficult to take photos. One only needs to find that picture of a camera on the phone. And then it’s just to snap away”. Kenneth described a similar experience:
Researcher:You were asked to take photos, what did you think about that?
Kenneth:Yes, that was super.
Researcher:Why?
Kenneth:No, because I often take photos anyway. It was totally natural for me.

Most participants expressed that they took the photos they wished to take and that the photos represented their travel to and/or from work. Still, they expressed difficulties connected to photovoice. During the interviews, comments were made regarding the quality of the photos. Ellen expressed that she found it difficult and said, “I’m not that very good at taking photos”. Kenneth, on the other hand, started talking about a photo he had taken and suddenly said: “No, but it is… Now when I look at it (the photo) in hindsight, I find it quite nice”.

Environmental challenges were mentioned by some of the participants. This included for instance that it was too dark to take photos, it was raining or that people were standing in the way of the object to be photographed. When Tobias showed his photos, he commented on the weather affecting the quality of a photo: “But most of them (the photos), most of them are easy to see. Apart from that one. Eh, it’s raining and the only thing we can get out of it [the photo] is that the house is probably white”. Regulations regarding data protection prevented the participants from taking photos of persons who had not given formal consent to be part of the study. This was reflected in the results. For instance, Dag said: “I tried to not take photos of people. Because a girl sat next to me, and when I tried to aim (the smartphone) at the vending machine, she got in the photo. So, I had to get up and stand in front of the vending machine to be able to take the photo”. Another participant also mentioned that he tried to avoid getting other persons in the photos, something that could result in embarrassing situations. Nicolai said “... it looks a bit stupid to stand and take photos of the floor, right. I’m thinking hell, what if someone sees that I take a photo and then they wonder, what the hell is he up to?”. In addition, one participant would have preferred to use a camera instead of a smartphone. 

Apart from environmental challenges, there were cognitive challenges. For instance, two of the participants forgot to take photos and some were too tired to engage in the activity. Some participants, therefore, had to give it a new try the day after. When Olav explained how it felt to participate in the photovoice he said: “It was great, but I didn’t take any photos yesterday because I was so tired that I had to go to bed, because I was a bit more tired than I expected so I didn’t manage to take any photos yesterday”. 

### 3.2. Sense of Self-Determination

Participation through photovoice was experienced to be open and with few boundaries. The participants mentioned a freedom of choice when it came to taking photos and that they could provide contextual as well as individual contributions. As Dag said: “I think that it was fun that I could take photos of whatever I wanted in a sense. Because you didn’t put any boundaries on what I could do.” The freedom to decide was also mentioned by Olav. He said: “Well because then I can talk a bit about which photos I like and which type of photos I don’t like. I could have taken photos I don’t like but then I had simply deleted them just as fast, at once.” The photovoice activity made it possible to contribute with contextual insights. Several of the participants took photos of their journey to and from work and contributed with insights to the innovation process. This included photos of places that were important to them, the scenery and settings that were stressful. Figure 2 shows examples of photos that described the context of the participants.

Through the photos and the interview, the participants also shared individual stories about themselves. For instance, Emma took a photo of the door to her apartment and explained why. 

Emma:That’s the photo of my door. In the apartment.

Researcher:Uhm.

Emma:Eh, there’s a bathroom and a sleeping room and a living room and a kitchen.

Researcher:Yes, would you like to tell me why you took the photo?

Emma:Uhm, why I took the photo. I’ve moved in.

Researcher:Was that something that made you feel happy or…

Emma:Eh, I’m so happy.

### 3.3. Sense of Ownership

The analysis of the data showed that the participants experienced that photovoice led to a sense of ownership. The sense of ownership was characterized by two sub-themes: signs of engagement and a sense of meaningfulness. The findings show that the participants were engaged, showed interest and were motivated during the photovoice activity. For instance, towards the end of the interview, Tobias asked: “Do you (the second author) wants me to take more photos? Just in case you need more photos”. The participants also expressed interest in further participation. For example, when asked if they would participate again, all participants expressed an interest in further research activities. While taking photos was a vital part of the experience, Nicolai stated, “participating through photovoice was fun and exciting… and talking (the interview) was possibly my favorite part of participating”. Nicolai elaborated and noted that he valued the opportunity to be part of something new: “it was exciting to be part of something new, and fun as well”.

In addition, the analysis showed that the participants experienced a sense of meaningfulness when participating through photovoice. They explained that participation was an opportunity to contribute to something important and meaningful. Several of the participants noted that they hoped that the results of their participation could contribute to helping other individuals with intellectual disabilities. Nicolai stated, “I hope (my participation) can lead to solutions that can help and make things easier for others as well”. In addition, photovoice made it possible for the participants to give descriptions of their lives and lived experiences. When asked about taking photos, Kenneth stated, “Taking photos was fine… it is an opportunity to show the wide spectrum (of the lives of people with intellectual disabilities)”. Moreover, participation was an opportunity to socialize and meet new people. The participants expressed that, when participating through photovoice, they could express themselves as they wished and without interruptions. Dag explained, “It is important to get the opportunity to talk to someone without being constantly interrupted. It is not easy to have a conversation when someone constantly interrupts”.

## 4. Discussion

In this section, we reflect on the process of using photovoice to empower participants with intellectual disability and understand user needs in an innovation process.

### 4.1. Reflections on Empowerment

One important element of user involvement, and photovoice in particular, is to contribute to empowerment of the participants [1,2,13,14]. Participants should be given the opportunity to select the photos they wish and to use these to guide the discussion on user needs. In addition, participants should be given the opportunity to contextualize the photos and tell a story connected to the photos [14]. In our study, the participants with intellectual disabilities did express a sense of self-determination and autonomy in connection to selecting which photos to take. For instance, they felt free to remove photos but also to discuss photos that were missing for different reasons. This is in line with Povee et al. [11], who found that photovoice can promote and facilitate empowerment by offering people with intellectual disabilities opportunities to make decisions and exert control over the process. In addition, our participants were encouraged to describe and express their opinions during the photovoice interview. They expressed a sense of control of how they wished to portray their experiences and elaborated on issues not directly connected to transportation.

Empowerment is characterized by and reflected in the ability to master issues that are of importance [25]. In our study, the participants expressed that they coped with and mastered the photovoice activity. They were engaged and motivated to take photos, talk about the photos and described photovoice as natural and fun. While people with intellectual disabilities tend to have difficulties maintaining motivation and engagement [16], photovoice can offer a more exciting way to engage in research [11]. This is in line with our findings, as the participants described the process as an enjoyable experience and expressed a desire to take more photos. The participants were also interested in the quality of the photos and some expressed a desire to take “better” photos. Another element of empowerment is the opportunity to engage in issues that concern and affect oneself [25]. A previous study found that photovoice may foster a sense of ownership as participants are involved in collecting data and are given the opportunity to shape and make policies relevant [25]. This was partly shown in our study as the participants expressed a sense of meaningfulness and ownership. The participants viewed photovoice as an opportunity to contribute to something that could lead to a solution that could help themselves and others to travel independently. Thus, the process of empowerment and the possibility of influencing the environment [25] was articulated in the user needs that influenced the design of the prototype. 

### 4.2. Reflections on Understanding User Needs

The gathering of insights in our study was somewhat different from previous studies involving participants with intellectual disability in photovoice [11,12]. In their process, participants collectively identified a theme to be explored and studied through the use of photovoice, and as a final step, the insights were communicated to policymakers [11,12]. Our focus of research, assistive technology to support independent transport, had been identified prior to the involvement of the photovoice participants. The theme was identified through participant observations of other persons with intellectual disabilities and through focus group interviews with representatives of support networks (parents, employers, nurses etc.). In addition, the target of our insights was not policymakers [11,12] but was used to design a prototype for assistive technology. The insights gathered through our photovoice process highlighted important elements of independent transport. Based on these insights, we designed a prototype of an assistive technology that supports reminders, time management and communication.

This shows that photovoice can contribute to user insights that can be a source for an innovation process and design of a prototype. When people with intellectual disabilities are allowed to express their interests, researchers are given the opportunity to understand the views and needs from their perspective rather than the perspective of caregivers, parents or other professionals [11,29]. According to Jurkowski [29], the engagement of people with intellectual disabilities in photovoice can improve validity as it fosters an authentic perspective of their experiences and knowledge. As a result, findings can be used to develop more effective programs [29], or as in our case, design and development of assistive technology.

### 4.3. Reflections on the Use of Photovoice

Photovoice offers an alternative approach for understanding user needs [13]. While it has been described as an opportunity to include participants who lack verbal fluency by sharing and presenting experiences visually [12], elements of photovoice have also been questioned, for instance, if a subsequent interview provides useful information and reflections of photos when involving participants with intellectual disabilities [30]. It has been suggested that it might be more beneficial to conduct interviews in the “field” as the photos are taken [30]. Nevertheless, in our study, the participants did reflect on the photos and did return to the photos without force to give examples or elaborate on their reasoning during the interviews. Our findings suggest that photovoice facilitated the understanding of their contextual and individual experiences and that the participants could make decisions based on their preferences. A similar finding was reported by Povee et al. [11], who state that photovoice captures the viewpoints and social realities of people with intellectual disabilities. As photovoice offers tangible representations of concepts and issues, it corresponds well with the concrete way of learning that fits people with intellectual disabilities [12]. Nevertheless, the ability to independently take photos varied among our participants. For one participant, a more guided approach as suggested by Overmars-Marx et al. [30] where the participant takes photos together with a researcher would probably have been more suitable.

We also found that the situation under study, in our case transport to work, took place in a context that made photovoice challenging for some participants. For instance, early mornings can be cognitively challenging in general. Timing of the study, both regarding day and time of the day, therefore, needs to be adjusted to the individual participant. Most participants described taking photos as “natural” and that it was something that they were used to in daily life. This reflects the use of social media in general and shows that people with intellectual disabilities should not be underestimated when contributing to an innovation process and technology design. In addition, photovoice might be more engaging and accessible as it does not require the ability to read and write [13]. Both these issues show the importance of preparation in regard to providing information to the participants and understanding the individual needs and abilities of the participants.

### 4.4. Limitations

There are potential limitations of our study that need to be considered. Although the number of interviews and participants is deemed sufficient by Braun and Clarke [31], this study had a limited number of participants and only two females. The number of participants and gender disparity may have contributed to the outcomes presented in this study. However, the participants represented the intended end user group of the assistive technology and used varied means of transportation, including maxitaxi, bus, subway and taxi. Still, it is important to acknowledge that people with intellectual disabilities are not a homogenous group. Our study was tailored to participants with certain abilities and they were able to articulate their experience of participating in photovoice. With this in mind, it is likely that the participants in this study were more skilled and capable of using technology than other people with intellectual disabilities. However, other studies such as the one by Cluley [32] include participants with mild, moderate and profound and multiple learning disabilities. Their results show that for participants with profound and multiple learning disabilities, a more flexible approach is recommended which includes both people with learning disabilities and representatives of the support network.

## 5. Conclusions

There is a need to engage disadvantaged user groups in innovation as it can result in improved digital solutions and inclusion of user groups and needs that have previously been overlooked. However, this calls for processes that are creative and empowering and which facilitate user involvement of groups such as people with intellectual disabilities. The findings in our study suggest that photovoice is a method that can contribute to sharing of contextual and individual needs and an empowerment process characterized by coping, self-determination and ownership. As photovoice emphasizes the visual capacity of people with intellectual disabilities, it promotes a relational perspective of intellectual disability. Photovoice can be a tool to reduce or remove some of the challenges that researchers might face when identifying user needs in design processes and a way to strengthen the individual’s capacity to cope with the demands of participating in an innovation and design process.

## Figures and Tables

**Figure 1 life-10-00272-f001:**
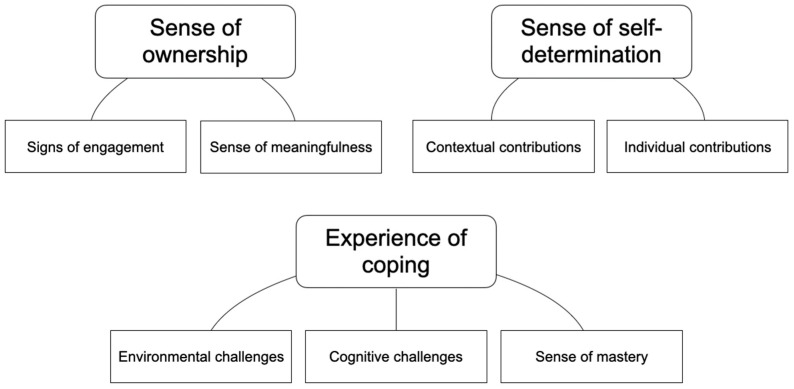
Thematic map showing the main themes and subthemes.

**Figure 2 life-10-00272-f002:**
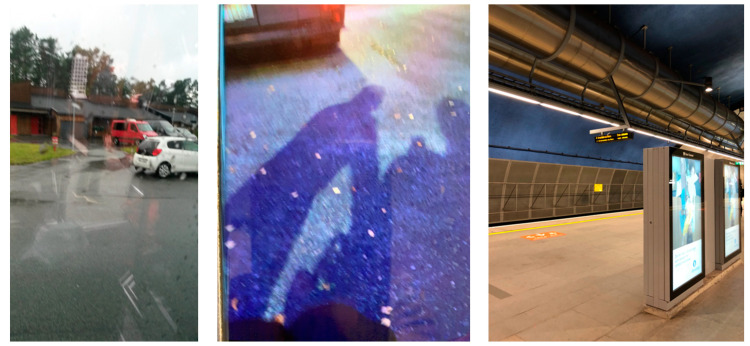
Example of participants’ photos describing the scenery and context of transport.

**Table 1 life-10-00272-t001:** Overview of the participants, with fictive names.

Participants	Gender	Transport	Number of Photos
Emma	Female	Maxitaxi	1
Tobias	Male	Maxitaxi	10
Ellen	Female	Bus	4
Olav	Male	Bus	6
Even	Male	Bus	0
Dag	Male	Taxi	2
Nicolai	Male	Subway + bus	3
Kenneth	Male	Taxi + bus	3
Adam	Male	Subway + bus	0

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
