# Peer review of "Photovoice—Towards Engaging and Empowering People with Intellectual Disabilities in Innovation"

_life, 2020, doi:10.3390/life10110272_

Round 1

Reviewer 1 Report

The manuscript is overall well written.

  1. What is missing is a central research question. The question The aim of our study is to report and reflect on the process of using photovoice as a means to understand user needs and to empower participants with intellectual disability in an innovation process (lines 61 and 62) should be followed by one or several specific research questions.
  2. This makes it easier to understand why this instruction On your way to and from work take photos of things, places or animals that are important, difficult or which makes you happy/calm/insecure/sad/scared/stressed 116 and 117) was given.
  3. Further why, connected to the central research question, was the interview shaped in this way? During the first part of the interview, the participants were asked to tell us about their current mode of transportation and their experiences connected to travelling. As a second part, the participants were asked to show us the photos they had taken, one by one (127-129). As you were not interested primarily in clients experiences concerning public transport but you were mainly interested in their photovoice experiences does the first part of the interview not mislead the participants concerning the purpose of this study?
  4. Why were the themes found in the thematic analysis significant with respect to your research aim and did the analysis include all material used or did you "cherry-pick" your results?

Reviewer 2 Report

This article makes a useful and well needed contribution to research and practice concerning the involvement of people with intellectual disabilities.  It is well written and the structure is clear.  However, I have a number of concerns. 

My main concern is that the paper does not fit with the aims and scope of the journal.  While I think the article is topical and provides practical insight into the use of photovoice as a means of engaging people with intellectual disabilities in innovation, something that I think is well needed, I wonder why the authors chose this journal for submission?  I would have think a journal focused on innovation or disability studies would provide a better fit?  If it was submitted to a journal focused on one of these issues, I think the important points made in the paper would reach a wider audience and have more impact on research and practice.  Life, with its biological focus, seems a strange choice.  I think ultimately this is a decision that should be made by the editor rather than the reviewer so I will leave this decision to them rather than rejecting the paper based on scope. However, I would encourage the authors to submit elsewhere and the editors to consider the article content against the journal scope.

If the article continues to be considered for publication in this journal I strongly recommend the following minor improvements are made.  I don't think the authors need to add an awful lot to include my suggestions but I have recommended some further reading to support with this.

1. The paper is based on an example that involved only nine participants.  The authors do reflect on this but I think more insight into the limitations this creates is needed.  Intellectual disability is not a homogeneous experience; rather it can be experienced very differently depending on both severity and contextual factors such as support networks, cultural factors, socio-economic position etc.  For this reason, reflecting on the involvement of nine participants is unlikely to provide a holistic insight into the involvement of people with intellectual disabilities in innovation.  I recognise that photovoice is an action research method and that action research is generally not conducted on a large scale, however, owing to the heterogeneity of participant focus, I think it is necessary to acknowledge this as a weakness.  While the authors do reflect on this, I think  more is needed.  I particularly think the authors need to reflect more on the fact that participants were recruited from sheltered workplaces and that the recruitment criteria explicitly focused on ability (those who can take photos with minimal assistance, those who can use public transport, those who can participate verbally).  This all means that the participants involved represented a particular ability/capacity level.   People with intellectual disabilities who can articulate their experiences verbally, who can use public transport and who can take photographs are very different from people who have a severe or profound intellectual disability.  I recommend reading Cluley’s article on including people with profound and multiple learning disabilities in photovoice - Cluley, V., 2017. Using photovoice to include people with profound and multiple learning disabilities in inclusive research. British Journal of Learning Disabilities45(1), pp.39-46.

In addition to the above, I also think the authors need to acknowledge the gender disparity between their participants and the impact this might have on the findings.  This is an assumption but, men and women may experience safety concerns associated with independent transport, differently.

2. The section discussing intellectual would benefit from a short acknowledgement of the growing role of critical disability studies and the move towards new materialist/post-human approaches to disability.  I recommend reading the articles below to support this.

Goodley, D., Lawthom, R. and Cole, K.R., 2014. Posthuman disability studies. Subjectivity7(4), pp.342-361.

Cluley, V., Fyson, R. and Pilnick, A., 2020. Theorising disability: a practical and representative ontology of learning disability. Disability & Society35(2), pp.235-257.

3. In the section discussing empowerment it would be useful to add a sentence or two linking empowerment to inclusive research.  I recommend reading:

Strnadová, I. and Walmsley, J., 2018. Peer‐reviewed articles on inclusive research: Do co‐researchers with intellectual disabilities have a voice? Journal of Applied Research in Intellectual Disabilities, 31(1), pp.132-141.

Walmsley, J. 2001. Normalisation, Emancipatory Research and Inclusive Research in Learning Disability, Disability & Society, 16,(2), pp., 187-205

4. The authors discuss the ethical considerations associated with consent and image content.  I think this requires slightly more consideration.  I recommend reading Claudia Mitchell’s discussing of ethics and visual research to support this.

Mitchell, C. 2011. Doing visual research. New York, NY and London, UK: Sage.

5. The data analysis section requires some expansion.  I wonder why thematic analysis was chosen when there are specific methods of analysis for photovoice?  It would be useful for the authors to at least acknowledge these methods, particularly interpretive engagement.  I recommend reading the articles below for more insight into this.

Cluley, V., Pilnick, A. and Fyson, R., 2020. Improving the inclusivity and credibility of visual research: interpretive engagement as a route to including the voices of people with learning disabilities in analysis. Visual Studies, pp.1-13.

Drew, S. and Guillemin, M., 2014. From photographs to findings: Visual meaning-making and interpretive engagement in the analysis of participant-generated images. Visual Studies29(1), pp.54-67.
